# Prohexadione Calcium Improves Rice Yield Under Salt Stress by Regulating Source–Sink Relationships During the Filling Period

**DOI:** 10.3390/plants14020211

**Published:** 2025-01-13

**Authors:** Rui Deng, Dianfeng Zheng, Naijie Feng, Aaqil Khan, Jianqin Zhang, Zhiyuan Sun, Jiahuan Li, Jian Xiong, Linchong Ding, Xiaohui Yang, Zihui Huang, Yuecen Liao

**Affiliations:** 1College of Coastal Agriculture Sciences, Guangdong Ocean University, Zhanjiang 524088, China; 2South China Center of National Saline-Tolerant Rice Technology Innovation Center, Zhanjiang 524088, China; 3Shenzhen Research Institute of Guangdong Ocean University, Shenzhen 518108, China

**Keywords:** rice, salt stress, prohexadione calcium, source–sink relationship, yield

## Abstract

Salt stress is an important factor affecting the growth and development of rice, and prohexadione calcium (Pro-Ca) plays an important role in alleviating rice salt stress and improving rice yield. However, there are few studies on how Pro-Ca improves rice yield under salt stress by regulating the source–sink metabolism. In this study, we used Guanghong 3 (salt-tolerant variety) and Huanghuazhan (salt-sensitive variety) as experimental materials to investigate the dynamic changes in the synthesis and partitioning of nonstructural carbohydrates among source–sink, the dynamic changes in related enzyme activities, the effects of the source–sink metabolism on yield in rice under salt stress and the effect of Pro-Ca during the filling period. The results of this study showed that Pro-Ca improved photosynthetic efficiency by increasing leaf photosynthetic gas exchange parameters and other stomatal factors on the one hand and, on the other hand, promoted sugar catabolism and reduced sugar synthesis by increasing leaf sucrose synthase activity and decreasing sucrose phosphate synthase activity, alleviating the inhibitory effect of high concentrations of sugars in the leaves on photosynthesis. Meanwhile, Pro-Ca promotes the transport of sugars from source (leaves) to sink (seeds), increases the sugar content in the seeds, and promotes starch synthesis in the seeds by increasing starch phosphorylase, which promotes seed filling, thus increasing the number of solid grains on the primary and secondary branches of the panicle in rice, increasing the 1000-grain weight, and ultimately increasing the seed setting rate and yield. These results indicated that Pro-Ca alleviated the inhibitory effect of salt stress on rice leaf photosynthesis through stomatal and non-stomatal factors. Meanwhile, Pro-Ca promotes the transport of rice sugars from source to sink under salt stress, regulates the source–sink relationship during the filling period of rice, promotes starch synthesis, and ultimately improves rice yield.

## 1. Background

Increasing urbanization and saltwater intrusion continue to reduce the amount of usable arable land in coastal areas, with approximately one billion hectares of land globally affected by salinization [1]. Saline and alkaline land is a scarce reserve land resource with great potential for comprehensive development, and the development and utilization of saline and alkaline land to improve saline and alkaline rice yield is the key to guaranteeing food security [2]. Rice (*Oryza sativa* L.), the staple food for about more than half of the world’s population, is considered a relatively salt-sensitive crop [3]. Salt stress inhibits the normal growth and development of rice plants and negatively affects grain yield traits, including grain yield, fruit set, 1000-seed weight, the effective number of panicles, panicle length, and number of grains per panicle [4].

Photosynthesis is the most important physicochemical process for energy production in higher plants and is very sensitive to salt stress [5]. Salt stress disrupts the lamellar structure of chloroplast granules and reduces the content of chlorophyll and carotenoids in leaves [6,7]. Under salt stress, the inhibition of plant leaf photosynthesis greatly reduces assimilate accumulation, accelerates leaf senescence and source-to-sink filling efficiency, and reduces rice yield [8]. Rice quality and yield depend mainly on the activities of sources, reservoirs, and streams and their interactions [9]. Sources absorb CO_2_ from ambient air, synthesize it into sugars, and transport the sugars to sinks [10]. Grains are considered to be major sink organs, and grain yield is determined by source–sink relationships [11]. Several researchers have found that low utilization of soluble carbohydrates is a major factor contributing to delayed grain growth [12,13]. Altering the availability of soluble carbohydrates showed different effects on grain filling in different types of rice. Soluble sugar is an important osmoregulatory substance used for storage and transfer of energy in plants, which includes monosaccharides and oligosaccharides such as glucose, fructose, and sucrose [14,15]. Plants can regulate the balance of substances in plants to resist adversity stress by changing the soluble sugar content [14]. The main nonstructural carbohydrates in rice seeds are soluble sugars and starch. Sucrose is the main form of sugar transported in rice and is produced by photosynthetic source tissues such as leaves and then transported to the reservoir tissues (seeds) through the phloem via the ectoplasmic or commensal pathway. Sucrose is then degraded to hexose, which is then metabolized by several metabolic and biosynthetic processes that ultimately lead to the synthesis of starch [16,17,18]. The expansion pressure difference caused by the osmotic potential of the “source” and “sink” continuously drives the transportation of sucrose, so the ratio of soluble sugar content to starch content of seeds not only reflects the output strength of the "source" but also determines the capacity of the “sink” for sucrose conversion and starch synthesis, which ultimately constrains the increase in crop yield [18]. The synthesis and accumulation of sucrose are mainly accomplished by sucrose synthase (SS) and sucrose phosphate synthase (SPS) [19]. Sucrose synthase utilizes UDPG, ADPG, and GDPG as glucose donors to react with fructose to produce sucrose [20]. Sucrose phosphate synthase is the rate-limiting enzyme for sucrose synthesis, which transfers glucose from UDPG to fructose 6-phosphate to synthesize sucrose phosphate, which is then catalyzed by sucrose phosphate phosphatase to produce sucrose [21,22,23]. Sucrose phosphate synthase activity is higher in photosynthetic tissues, which facilitates the supply of photosynthetic products to plant growth [24]. Sucrose synthase activity was significantly and positively correlated with the rate of starch accumulation. The grouting process of rice is mainly the process of seed starch synthesis and accumulation, and this process mainly relies on the synergism of various enzymatic reactions. In addition, starch phosphorylase is the key enzyme of rice starch synthesis and plays an important role in regulating the starch synthesis and accumulation in rice [25].

Previous studies have found that exogenous plant growth regulators can increase the resistance of rice to salt stress [26]. Prohexadione calcium (Pro-Ca), a naturally derived cyclohexanecarboxylic acid, is a class of exogenous plant growth regulators that delay plant growth and inhibit gibberellin synthesis in plants [27]. Pro-Ca improves crop seed yield and salt tolerance and has no residual toxicity to crops [28]. Previous studies showed that moderate amounts of Pro-Ca could shorten the stem height of rice and increase the content of photosynthetic pigments and the net photosynthetic rate of plants [29]. The chlorophyll fluorescence index, Fv/Fm, is the maximum photochemical quantum yield of PS II and can be used as a photoinhibition index to reflect the efficiency of light energy conversion in the active center of PS II [30]. In addition, the potential photochemical activity of PS II (Fv/Fo) is another important photochemical parameter that can be used to assess leaf photosynthetic capacity [31,32]. Previous studies have demonstrated that moderate amounts of Pro-Ca increase Fv/Fm and Fv/Fo and maintain the activity of photosystem II (PSII), which improves photosynthetic efficiency and promotes crop growth [3]. The 5–7 days before the rice breakthrough stage is the best period for the control of rice pests and diseases in agricultural production, and it is also a critical period for rice management. Whether or not the prevention and control are proper and the management is appropriate is directly related to the late rice yield.

Previous studies have mostly focused on exploring the antioxidant mechanism of Pro-Ca to alleviate salt stress in rice. However, there are fewer studies on how Pro-Ca improves rice yield under salt stress by regulating the source–sink metabolism. In this experiment, we used the salt-tolerant rice Guanghong 3 (GH3) and the salt-sensitive variety Huanghuazhan (HHZ) as experimental materials to investigate the mechanisms of Pro-Ca that affect the regulation of chlorophyll content, photosynthesis, and nonstructural carbohydrate content of leaves (source organ) and seeds (sink organ), the activities of related enzymes (SS, SPS, SP, etc.), and the effect of source and sink material partitioning on yield in rice under salt stress. This study helps us investigate the dynamic changes in the synthesis and allocation of nonstructural carbohydrates between the source and sink, the dynamic changes in the activities of related enzymes, and the effects of source and reservoir metabolism on yield during the whole filling period of rice under salt stress, as well as the effect of Pro-Ca.

## 2. Results

### 2.1. Effect of Pro-Ca on Phenogenesis in Rice Under Salt Stress

Salt stress negatively affects the aboveground and belowground parts of rice, and Pro-Ca has an alleviating effect (Figure 1).

On day 15 of the filling stage, the S treatment decreased the leaf fresh weight by 6.73% and 18.41%, and the leaf dry weight by 8.41% and 29.35%, respectively for GH3 and HHZ, compared with CK (Figure 2A,B). Compared with the S treatment, the leaf fresh weight was increased by 3.94% and 21.64%, and the leaf dry weight was significantly decreased by 6% and 27.61% in GH3 and HHZ, respectively, by the SP treatment. These changes reached a significant level only in the HHZ variety (Figure 2A,B). There was no significant change in the root system of GH3 under salt stress, and the root system of HHZ was significantly inhibited. The root fresh weight and dry weight of S-treated HHZ were significantly reduced by 27.68% and 45.77%, respectively, compared with CK-treated HHZ. The root fresh and dry weights of SP-treated HHZ increased by 31.34% and 44.62%, respectively, compared to S-treated rice (Figure 2C,D).

Salt stress inhibited panicle as well as internode growth in both rice varieties (Figure 3). On day 15 of the filling stage, the S treatment significantly decreased the panicle fresh weight of the main stem by 23.20% and 27.61%, the panicle fresh weight of the tiller stem by 24.25% and 69.31%, the panicle dry weight of the main stem by 21.93% and 33.33%, the panicle dry weight of the tiller stem by 38.96% and 73.43%, the length of internode under the panicle by 10.38% and 16.05%, and the diameter of the internode under the panicle by 20.78% and 34.01%, respectively, for GH3 and HHZ, compared with the CK. Compared to the S treatment, the SP treatments increased the panicle fresh weight of the main stem by 23.49% and 30.74%, the panicle fresh weight of the tiller stem by 18.98% and 191.61%, the panicle dry weight of the main stem by 19.52% and 30.63%, the panicle dry weight of the tiller stem by 14.57% and 238.16%, the length of the internode under the panicle by 28.70% and 9.08%, and the diameter of the internode under the panicle by 17.33% and 25.43%, respectively, for GH3 and HHZ.

### 2.2. Effect of Pro-Ca on Photosynthetic Pigments and Gas Exchange Parameters of Rice Flag Leaves (Inverted Leaf) at the Filling Stage Under Salt Stress

Under salt stress, chlorophyll synthesis was blocked in rice leaves (Figure 4A–D). For S treatments, there were significantly decreasing trends of 6.09% and 13.71% in chlorophyll a content, 5.04% and 36.65% in chlorophyll b content, 14.47% and 7.39% in carotenoid content, and 5.72% and 21.56% in total chlorophyll content, respectively, for GH3 and HHZ, compared with CK treatments. Compared to the S treatment, the SP treatment increased chlorophyll a content by 7.19% and 23.17%, chlorophyll b content by 15.99% and 67.89%, carotenoid content by 9.74% and 7.23%, and total chlorophyll content by 10.35% and 35.53%, respectively, for GH3 and HHZ.

Salt stress significantly reduced the photosynthetic gas exchange parameters of GH3 and HHZ (Figure 4E–H). the S treatment significantly decreased Pn by 39.22% and 35.53%, Ci by 6.46% and 9.93%, Gs by 55.77% and 58.49%, and Tr by 48.03% and 45.39%, respectively for GH3 and HHZ, compared with CK. Compared to the S treatment, the SP treatment significantly increased Pn by 26.22% and 45.88%, Ci by 6.83% and 7.52%, Gs by 65.22% and 131.82%, and Tr by 47.27% and 59.56%, respectively, for GH3 and HHZ under salt stress.

### 2.3. Effect of Pro-Ca on Nonstructural Carbohydrates of Rice Leaves Under Salt Stress

The amounts of sucrose, fructose, and total soluble sugar in the leaves of both types of rice slowly rose and then fell during the filling stage (Figure 5). The leaves of GH3 had the highest sugar content on the 10th day, while the leaves of HHZ had the highest on the 15th day. Sucrose content, fructose content, and soluble sugar content in the leaves of both rice varieties increased to certain levels under salt stress but decreased after the application of the Pro-Ca treatment. Compared to CK, the S treatment significantly increased the sucrose content in leaves of GH3 and HHZ by 8.57–29.40% and 3.12–39.35%, respectively, during the filling period under salt stress. Compared to the S treatment, the SP treatment significantly decreased the sucrose content in the leaves of GH3 and HHZ by 14.07–29.36% and 8.01–25.56%, respectively, during the filling period. Compared to CK, the S treatment significantly increased the fructose content in the leaves of GH3 and HHZ by 13.64–33.69% and 8.63–69.35%, respectively, during the filling period under salt stress. The fructose content in the leaves of GH3 and HHZ significantly decreased by 1.53–33.19% and 3.54–17.98%, respectively, during the filling period under the SP treatment, compared to the S treatment. The soluble sugar content in the leaves of GH3 and HHZ significantly increased by 17.40–43.38% and 11.74–34.20%, respectively, during the filling period under the S treatment, compared to the CK treatment. The soluble sugar content in the leaves of GH3 and HHZ decreased by 2.85–12.94% and 2.15–11.67%, respectively, during the filling period under the SP treatment, compared to the S treatment.

### 2.4. Effect of Pro-Ca on the Activities of Sugar-Related Enzymes in Rice Leaves Under Salt Stress

The SS activity of rice leaves showed a curvilinear change of increasing and then decreasing, and the SS activity of GH3 and HHZ leaves peaked on the 10th and 15th day of the filling period, respectively (Figure 6A,B). Compared to CK, the S treatment significantly decreased the SS activity in the leaves of both varieties during the filling period. There were corresponding percentage range decreases of 16.73% to 38.85% in GH3 and 25.06% to 43.28% in HHZ under the S treatment. Compared to S, the SP treatment significantly increased the SS activity in the leaves of both varieties during the filling period. There were corresponding percentage range increases of 10.62% to 17.05% in GH3 and 9.69% to 21.12% in HHZ under the S treatment.

The SPS activity of rice leaves showed a curvilinear change of increasing and then decreasing, and the sucrose synthase activity of GH3 and HHZ leaves peaked on the 10th and 15th days of the filling period, respectively (Figure 6C,D). Compared to CK, the S treatment significantly increased the SPS activity in the leaves of both varieties during the filling period. There were corresponding percentage range increases of 25.96% to 40.74% in GH3 and 23.83% to 44.56% in HHZ, respectively, under the S treatment. Compared to S, the SP treatment significantly decreased the SPS activity in the leaves of both varieties during the filling period. There were corresponding percentage range decreases of 8.68% to 19.32% in GH3 and 10.23% to 17.24% in HHZ, respectively, under the S treatment.

### 2.5. Effect of Pro-Ca on Nonstructural Carbohydrate Content of Rice Seeds Under Salt Stress

During the filling stage, the sucrose content, fructose content, and total soluble sugar content in the seeds of both rice varieties showed a slight increase, which peaked on day 10 and then continued to decline (Figure 7). Under salt stress, the sucrose content, fructose content, and soluble sugar content in the seeds of the two rice varieties showed different degrees of decline; the sucrose content, fructose content, and soluble sugar content increased after the Pro-Ca treatment, and the effect on HHZ was more significant. The sucrose content in the seeds of GH3 and HHZ decreased by 0.99–45.13% and 10.81–41.86%, respectively, during the filling period under the S treatment, compared to the CK treatment. Compared to the S treatment, the sucrose content in the seeds of GH3 and HHZ significantly increased by 7.89–61.02% and 5.57–45.60%, respectively, during the filling period under the SP treatment. Compared to the CK treatment, the fructose content in the seeds of GH3 and HHZ significantly decreased by 15.29–48.37% and 7.13–51.33%, respectively, during the filling period under the S treatment. The fructose content in the seeds of GH3 and HHZ significantly increased by 5.34–64.89% and 18.18–68.56%, respectively, during the filling period under the SP treatment, compared to the S treatment. The soluble sugar content in the seeds of GH3 and HHZ significantly decreased by 7.70–22.17% and 6.65–22.61%, respectively, during the filling period under the S treatment, compared to the CK treatment. The soluble sugar content in the seeds of GH3 and HHZ increased by 4.48–18.61% and 0.33–25.63%, respectively, during the filling period under the SP treatment, compared to the S treatment.

At the filling stage, the starch content of the seeds of both varieties showed an increasing trend. Under salt stress, the starch content in the kernels of both rice varieties decreased to different degrees, and the starch content increased after the Pro-Ca treatment (Figure 8). Compared to the CK treatment, the S treatment significantly decreased the starch content in the seeds of both varieties during the filling period. There were corresponding percentage range decreases of 8.29% to 11.20% in GH3 and 7.42% to 12.32% in HHZ under the S treatment. Compared to the S treatment, the SP treatment significantly increased the starch content in the seeds of both varieties during the filling period. There were corresponding percentage range increases of 3.72% to 6.11% in GH3 and 2.89% to 7.57% in HHZ under the S treatment.

### 2.6. Effect of Pro-Ca on Starch Phosphorylase Activity of Rice Seeds Under Salt Stress

Under salt stress, the starch phosphorylase activities in the seeds of both rice varieties showed a tendency to increase first, then decrease, and reached the peak on the tenth day of the grouting period (Figure 9). Compared to the CK treatment, the S treatment significantly decreased the SPS activity in the seeds of both varieties during the filling period (Figure 8). There were corresponding percentage range decreases of 9.39% to 19.97% in GH3 and 13.25% to 25.47% in HHZ under the S treatment. Compared to the S treatment, the SP treatment significantly increased the SPS activity in the seeds of both varieties during the filling period (Figure 8). There were corresponding percentage range increases of 6.08% to 17.79% in GH3 and 10.03% to 23.12% in HHZ under the S treatment.

### 2.7. Effects of Pro-Ca on Rice Yield and Its Components Under Salt Stress

Salt stress significantly reduced the yield components of both rice varieties (Table 1, Figure 10). In 2023, compared with CK-treated GH3, the length of the main stem panicle, number of solid grains on the primary branches, and number of solid grains on the secondary branches of S-treated GH3 decreased by 5.63%, 5.99%, and 27.91%, respectively. In addition, the number of primary branches and secondary branches did not change significantly; however, the number of empty grains on the primary branches and the number of empty grains on the secondary branches significantly increased by 31.17% and 62.77%, respectively. Compared with CK-treated HHZ, the length of the main stem panicle, number of solid grains on the primary branches, and number of solid grains on the secondary branches of S-treated HHZ were significantly reduced by 8.29%, 6.76%, and 43.33%, respectively; the number of primary branches and the number of secondary branches did not change significantly; and the number of empty grains on the primary peduncle and the number of empty grains on the secondary branches were significantly increased by 28.54% and 114.30%, respectively. The panicle length of the main stem, the number of solid grains on the primary branches, and the number of solid grains on the secondary branches of SP-treated GH3 increased by 5.77, 22.48, and 32.59%, respectively; the number of primary branches and the number of secondary branches did not change significantly; and the number of empty grains on the primary branches and the number of empty grains on the secondary branches were significantly reduced by 44.23 and 24.42%, respectively, when compared with S-treated GH3. The number of empty grains on the primary branches and the number of empty grains on the secondary branches were significantly reduced by 44.23 and 24.42%, respectively. The length of the main stem panicle, the number of solid grains on the primary branches, and the number of solid grains on the secondary branches were significantly increased by 10.08%, 35.73%, and 83.87%, respectively, in SP-treated HHZ compared to S-treated HHZ. The number of primary branches and secondary branches did not change significantly, and the number of empty grains on the primary branches and the number of empty grains on the secondary branches were significantly reduced by 31.87% and 25.03%, respectively. Compared with the CK treatment, GH3 and HHZ exhibited decreases of 2% and 9%, respectively, in the number of solid grains on the primary branches and decreases of 13% and 21%, respectively, in the number of solid grains on the secondary branches of the total number of grains in the whole panicle under the S treatment. Compared to the S treatment, the SP treatment increased the number of solid grains on the primary branches of GH2 and HHZ by 6% and 8%, respectively, of the total number of grains in the whole panicle. In addition, the number of solid grains on the secondary branches of GH3 and HHZ increased by 12% and 20%, respectively, of the total number of grains in the whole panicle under the SP treatment. Compared with the CK treatment, the S treatment increased the number of empty grains on the primary branches of GH3 and HHZ by 3% and 5%, respectively, of the total number of grains in the whole panicle and increased the number of empty grains on the secondary branches of GH3 and HHZ by 12% and 17%, respectively, of the total number of grains in the whole panicle. Compared to the S treatment, GH3 and HHZ showed a 5% and 5% reduction, respectively, in the number of empty grains on the primary branches and an 8% and 12% reduction, respectively, in the number of empty grains on the secondary branches of the total number of grains in the whole panicle under SP treatment. In 2024, the main stem panicle length, number of solid grains on the primary branches, and number of solid grains on the secondary branches decreased by 7.67%, 11.34%, and 27.61%, respectively; the number of primary branches and secondary branches did not change significantly; and the number of empty grains on the primary branches and the number of empty grains on secondary branches significantly increased by 24.86% and 36.38%, respectively, in S-treated GH3, compared to CK-treated GH3. Compared with CK-treated HHZ, the length of the main stem panicle, number of solid grains on the primary branches, and number of solid grains on the secondary branches of S-treated HHZ were significantly reduced by 7.10%, 26.32%, and 53.47%, respectively; the number of primary branches and secondary branches did not change significantly; and the number of empty grains on the primary branches and the number of empty grains on the secondary branches were significantly increased by 68.99% and 74.91%, respectively. Compared with the S treatment, the length of the main stem spike, number of solid grains on the primary branches, and number of solid grains on the secondary branches increased by 7.01%, 12.78%, and 31.71%, respectively; the number of primary branches and number of secondary branches did not change significantly; and the number of empty grains on the primary branches and the number of empty grains on the secondary branches were significantly reduced by 21.88% and 21.17%, respectively, in GH3 under the SP treatment. The length of the main stem panicle, number of solid grains on the primary branches, and number of solid grains on the secondary branches were significantly increased by 7.38%, 35.73%, and 83.87%, respectively; there was no significant change in the number of primary branches and the number of secondary branches; and the number of empty grains on the primary branches and the number of empty grains on the secondary branches were significantly reduced by 31.87% and 25.03%, respectively, in SP-treated HHZ, compared with S-treated HHZ. Compared with the CK treatment, under the S treatment, GH3 and HHZ exhibited a reduced number of solids on the primary branches by 3% and 7%, respectively, of the total number of grains in the whole panicle and a reduced number of solids on the secondary branches by 10% and 21%, respectively, of the total number of grains in the whole panicle. Compared with the S treatment, under the SP treatment, GH3 and HHZ exhibited an increased number of solid grains on the primary branches by 3% and 9%, respectively, of the total number of grains in the whole panicle and an increased number of solid grains on the secondary branches by 9% and 17%, respectively, of the total number of grains in the whole panicle. Compared with the CK treatment, the number of empty grains on the primary branches of GH3 and HHZ increased by 4% and 8%, respectively, of the total number of grains in the whole panicle, and the number of empty grains on the secondary branches increased by 10% and 16%, respectively, of the total number of grains in the whole panicle under the S treatment. Compared to the S treatment, the number of empty grains reduced on the primary branches of GH3 and HHZ accounted for 4% and 7%, respectively, of the total number of grains in the whole panicle, and the number of empty grains reduced on the secondary branches accounted for 8% and 11%, respectively, of the total number of grains in the whole panicle under the SP treatment.

Salt stress reduced the number of solid grains per panicle, 1000-grain weight, seed setting rate, and yield per plant in rice (Table 2, Figure 10). In 2023, the number of solid grains per panicle, 1000-grain weight, fruiting rate, and yield per plant in S-treated GH3 were significantly reduced by 26.26%, 11.59%, 18.06%, and 42.11%, respectively, compared with CK-treated GH3. The number of solid grains per panicle, 1000-grain weight, seed setting rate, and yield per plant were significantly reduced by 43.21%, 14.13%, 29.33%, and 52.40%, respectively, in S-treated HHZ compared to CK-treated HHZ. The number of solid grains per panicle, 1000-grain weight, fruiting rate, and yield per plant were significantly increased by 41.34%, 12.5%, 11.86%, and 53.61%, respectively, in SP-treated GH3 compared to S-treated GH3. The number of solid grains per panicle, 1000-grain weight, fruiting rate, and yield per plant were significantly increased by 37.91%, 14.98%, 18.87%, and 55.03%, respectively, in SP-treated HHZ compared to S-treated HHZ. In 2024, the number of solid grains per panicle, 1000-grain weight, fruiting rate, and yield per plant were significantly decreased by 14.14%, 12.49%, 17.74%, and 25.38%, respectively, in S-treated GH3 compared to CK-treated GH3. The number of solid grains per panicle, 1000-grain weight, fruiting rate, and yield per plant were significantly reduced by 31.62%, 17.93%, 21.88%, and 43.75%, respectively, in S-treated HHZ compared to CK-treated HHZ. The number of solid grains per panicle, 1000-grain weight, fruiting rate, and yield per plant were significantly reduced by 13.45%, 13.88%, 17.65%, and 29.91%, respectively, in SP-treated GH3 compared to S-treated GH3. The number of solid grains per panicle, 1000-grain weight, fruiting rate, and yield per plant were significantly reduced by 34.7%, 16.51%, 24%, and 59.46%, respectively, in SP-treated HHZ compared to S-treated HHZ.

### 2.8. Correlation Analysis

In order to identify the correlation between these traits more effectively, a Pearson correlation analysis was performed for 24 representative traits (Figure 11). The results revealed that the sucrose content, fructose content, and soluble sugar content in leaves showed a positive correlation. The sucrose content, fructose content, and soluble sugar content in seeds showed a positive correlation. The leaf sucrose content, leaf fructose content, and leaf soluble sugar content were negatively correlated with the number of solid grains in primary branches, number of solid grains in secondary branches, 1000-grain weight, seed setting rate, and yield and positively correlated with the number of empty grains on the primary branches and the number of empty grains on the secondary branches. The seed sucrose content, seed fructose content, and seed soluble sugar content were positively correlated with the number of grains on the primary branches, number of grains on the secondary branches, 1000-grain weight, seed setting rate, and yield and negatively correlated with the number of empty grains on the primary branches and the number of empty grains on the secondary branches. The number of solid grains on the primary branches and the number of solid grains on the secondary branches were negatively correlated with the number of empty grains on the primary branches and the number of empty grains on the secondary branches. There was a positive correlation among the number of solid grains on the primary branches, number of solid grains on the secondary branches, 1000-grain weight, fruiting rate, and yield per plant. There was a negative correlation among the number of empty grains on the primary branches, number of empty grains on the secondary branches, 1000-grain weight, seed setting rate, and yield per plant. The thousand-grain weight, seed setting rate, and yield per plant showed a positive correlation.

## 3. Discussion

### 3.1. Effect of Pro-Ca on Morphogenesis in Rice Under Salt Stress

Excessive salinity reduces the activity of meristematic tissue cells and interferes with normal physiological and biochemical processes, thus adversely affecting plant growth. Many studies have shown that the growth of the aboveground parts and root systems of plants is susceptible to inhibition in high salt environments, which ultimately reduces the accumulation of biomass [33]. In this study, we found that salt stress inhibited the growth of the HHZ root system and Pro-Ca alleviated the damage to the root system caused by salt stress, but the GH3 rice root system did not produce significant changes, which may be due to the fact that HHZ is more sensitive to salt stress, therefore causing the root system to be more susceptible to damage [34]. It also demonstrated that Pro-Ca was more effective in alleviating root damage in salt-sensitive varieties compared to salt-tolerant varieties. The leaf is an important source organ of rice, providing energy for rice growth and development [35]. In this study, both the dry and fresh weights of rice leaves decreased to different degrees under salt stress, and both the dry and fresh weights of rice leaves increased to different degrees after the application of Pro-Ca. This proves that salt stress has a negative effect on rice leaves (source organ), while Pro-Ca has a certain mitigating effect. We also found that Pro-Ca was more effective in alleviating leaf (source organ) damage in salt-sensitive varieties than in salt-tolerant varieties. Numerous studies have demonstrated that the longest internode connects the stem and panicle during the rice grain filling stage, controls the efficiency of water and nutrient transport from leaves (source organ) and stems to grains (sink organ), and influences grain filling by regulating energy transfer, which ultimately and seriously affects rice yield [36]. In this study, we found that the length of the internode under the panicle (longest internode) and the diameter of the internode under the panicle (longest internode) were significantly reduced under salt stress, and Pro-Ca significantly increased the length of the internode under the panicle (longest internode) and the diameter of the internode under the panicle, which improved the efficiency of water and nutrient transportation from the source to the reservoir, increased the rice grouting capacity, and, ultimately, increased the rice yield.

### 3.2. Effect of Pro-Ca on Photosynthesis in Leaves (Source Organs) Under Salt Stress

Photosynthesis is one of the most important processes in the chloroplasts of higher plants. Plants can convert solar energy into chemical energy through photosynthesis to provide energy for plant growth [35]. Previous studies have repeatedly demonstrated that salt stress affects rice photosynthesis through stomatal and non-stomatal factors, which, in turn, affects rice yield. In this experiment, the effects of salt stress and spraying Pro-Ca on leaf photosynthesis were investigated by determining the gas exchange parameters (Pn, Gs, Ci, and Tr) of the leaves. The results showed that all four indicators decreased under salt stress. The spraying of Pro-Ca under salt stress showed different increases in all four indicators [37]. The stomatal conductance (Gs) increased significantly, indicating that the inhibition of diffusion of CO_2_ from the environment to chloroplasts was reduced [3]. The increase in the photosynthetic rate may be due to the increase in the intracellular CO_2_ content as a result of stomatal opening. Studies have shown that non-stomatal factors also cause a decrease in Pn, including a large accumulation of Na^+^, a decrease in intracellular K^+^ content, a decrease in photosynthetic pigments, damage to the ultrastructure of chloroplasts, and a decrease in the activity of key photosynthetic enzymes [38]. In this study, the chlorophyll a, chlorophyll b, and carotenoid contents decreased. The chlorophyll a, chlorophyll b, and carotenoid contents increased after spraying Pro-Ca under salt stress, suggesting that non-stomatal factors influence photosynthetic efficiency. The abovementioned results indicate that both salt stresses increased the photosynthetic efficiency and alleviated the effects of salt stress on rice through stomatal and non-stomatal factors after spraying Pro-Ca under salt stress. Meanwhile, we also found that the photosynthetic gas exchange traits and photosynthetic pigment content of the salt-sensitive variety, HHZ, decreased more significantly under salt stress compared with the salt-tolerant variety, GH3, and the photosynthetic gas exchange parameters and photosynthetic pigment content of the salt-sensitive variety, HHZ, increased more significantly after the foliar application of Pro-Ca, which proved that the regulation of photosynthesis of salt-sensitive varieties was more significantly affected by Pro-Ca.

### 3.3. Effect of Pro-Ca on Source–Sink Metabolism Under Salt Stress

Salt stress may affect the assimilate partitioning process, reducing the ability of source organs (leaves) to export nonstructural carbohydrates to sink organs (seeds) and resulting in a shortage of nutrients required for seed filling, leading to lower yields [39,40,41]. The sucrose content, fructose content, and total soluble sugar content in the leaves of the two rice varieties in this study showed a gradual increase, followed by a gradual decrease, in seven sampling occasions, with the sugar content in the leaves of GH3 reaching a peak on the 10th day and that in the leaves of HHZ reaching a peak on the 15th day. This is because the activity of the enzymes related to the synthesis of sugars in the leaves was higher at the early stage of the filling period, and the rate of sugar synthesis in the leaves was greater than the rate of transfer to the seeds, so the sugar content in the leaves gradually increased. After reaching the peak, as the leaves senesced, the enzyme activity decreased, and the sugars were continuously transported to the seeds, so the sugar content in the leaves gradually decreased, which was consistent with the results of our related enzyme activity measurements. Sucrose is a product of photosynthetic assimilation in rice leaves and the main form of material transported to the spike for grain filling during the filling period [42]. The activities of key enzymes involved in the sucrose–starch conversion process in rice leaves determine the sugar content [43]. SPS and SS are two key enzymes that catalyze the synthesis and catabolism of sucrose in plants [19]. SPS catalyzes the formation of sucrose, while SS mainly catalyzes the catabolism of sucrose to produce fructose, and both jointly regulate the operation of the sucrose metabolism [44,45]. In this study, salt stress significantly increased SPS activity and decreased SS activity in leaves, and Pro-Ca appropriately decreased SPS and increased SS activity. This suggests that salt stress stimulates SPS activity, decreases SS activity, increases sucrose and fructose content, maintains osmotic balance, and contributes to the improvement of salt tolerance in rice. However, the significant amounts of sugars accumulated in plants under stress conditions also inhibit normal photosynthesis and delay plant growth [25,46,47]. Therefore, salt stress accumulates a large amount of sugar compared to CK, which further inhibits photosynthesis while relieving osmotic stress, thus limiting its growth through feedback regulation. In this experiment, Pro-Ca promoted sucrose catabolism by increasing leaf SS activity; meanwhile, it decreased SPS activity and reduced sucrose synthesis to alleviate the inhibition of photosynthesis by high concentrations of sucrose in the leaves [25]. A previous study found that the sucrose content, fructose content, and total soluble sugar content of the seeds showed a continuous decrease during the filling process [48]. In this study, the sucrose content, fructose content, and total soluble sugar content in the seeds of both rice varieties showed a slightly increasing trend, which peaked on the 10th day and then decreased continuously. This may be due to the late filling of some grains, where the rate of sucrose transport to the seeds was higher than the rate of sucrose utilization by the seeds. Compared with the CK treatment, the contents of soluble sugar, sucrose, and fructose in the seed grains decreased significantly under salt stress, and the contents of soluble sugar, sucrose, and fructose increased significantly after the Pro-Ca treatment. This may be due to the inhibition of soluble sugar, sucrose, and fructose transport from leaves to seeds under salt stress, while Pro-Ca alleviated the inhibition of material transport by salt stress and regulated the source–sink metabolism.

Starch makes up 80–85% of the dry weight of the grain, and starch is closely related to rice yield quality [49]. Photosynthetic products are first transported to the grain as sucrose through the bast and then stored as starch through a series of enzymatic reactions [42]. Previous studies have found that starch content shows a curvilinear increase during the filling period, and the results of the present study were consistent with the previous study [48]. Moreover, in this study, we found that the starch content of both rice varieties under salt stress decreased significantly compared with CK-treated rice, and the starch content increased significantly after Pro-Ca treatment. This indicates that, on the one hand, salt stress inhibits the transport of sucrose from source tissues (leaves) to sink tissues (grains), thus reducing the sucrose content in the seeds, and, on the other hand, it reduces the activity of enzymes related to starch synthesis, thus inhibiting starch synthesis and ultimately decreasing the yield of rice [18]. Pro-Ca enhances the activity of enzymes related to starch synthesis, alleviates the inhibitory effect of salt stress on rice grouting, increases rice starch content, and ultimately improves rice yield by facilitating the transport of sucrose from source tissues (leaves) to sink tissues (seeds). Starch phosphorylase is an essential enzyme in starch synthesis during rice endosperm development because the enzyme plays a key role in both the early and mature stages of starch granule formation [50]. In this study, the starch phosphorylase activities in the seeds of both rice varieties in different treatment groups showed a tendency to increase first, then decrease, and peak on the tenth day after flushing. Compared with the CK treatment, the starch phosphorylase activities of both rice varieties significantly decreased under the S treatment, and the starch phosphorylase activities of both rice varieties significantly increased after the SP treatment, which also proved our abovementioned inference that salt stress reduces the activities of enzymes related to starch synthesis by decreasing starch synthesis. Pro-Ca can moderately alleviate the negative effects of salt stress, promote starch synthesis, increase the starch content in the seeds, and ultimately improve rice yield. Meanwhile, we also found that the sugar content and starch phosphorylase activity of HHZ seeds increased more significantly under Pro-Ca than that of GH3 compared with salt stress, which proved that Pro-Ca had a more significant source–sink modulation effect on the salt-sensitive variety.

### 3.4. Effect of Pro-Ca on Organogenesis and Yield of Rice Reservoirs Under Salt Stress

Rice yield is the combined result of the joint determination of photosynthetic material production capacity and functional allocation of assimilates. Salinity affects rice growth, which in turn affects the yield components and yield of rice. Salt stress significantly affects rice yield traits, such as the seed setting rate, effective number of panicles, panicle length, 1000-grain weight, number of primary branches, number of secondary branches, number of solid grains on primary branches, number of empty grains on primary branches, number of solid grains on secondary branches, number of empty grains on secondary branches, and so on [51,52]. In this study, we found that salt stress significantly decreased the number of solid grains on the primary and secondary branches and increased the number of empty grains on the primary and secondary branches, and we found that the 1000-grain weight also decreased significantly under salt stress. This might be due to the fact that salt stress inhibits the transport of sugars from the source (leaves) to the reservoir (seeds) and reduces the activity of starch phosphorylase, which inhibits the synthesis of starch in the seeds and suppresses rice filling, thus leading to a decrease in the number of solid grains on the branching peduncle and 1000-grain weight, which ultimately reduces the seed setting rate and yield [51].

In this study, we found that Pro-Ca significantly increased the number of solid grains on the primary and secondary branches, decreased the number of empty grains on the primary and secondary branches, and significantly increased the 1000-grain weight. This might be due to the fact that Pro-Ca could promote the transport of sugars from the source to the sink under salt stress and increase the activity of starch phosphorylase, which promotes the synthesis of starch in the grain and protects the ability of the rice to fill, thus increasing the number of solid grains on the branching peduncle and 1000-grain weight and ultimately increasing the seed setting rate and yield of rice [53,54]. Most of the yield traits decreased more in HHZ, indicating that HHZ was more sensitive to the salt environment than GH3. Compared with the S treatment, the number of solid grains per panicle, 1000-grain weight, seed setting rate, and yield per plant increased more significantly in HHZ under the SP treatment than in GH3, which also proved that Pro-Ca was more effective in regulating salt-sensitive varieties.

## 4. Conclusions

Pro-Ca increased the photosynthetic efficiency of rice under salt stress through stomatal and non-stomatal factors, and the length, as well as the diameter, of the first internode under the panicle was also significantly increased. Salt stress significantly increased the SPS activity, decreased the SS activity, increased the sucrose, fructose, and soluble sugar content, and maintained the osmotic balance in the leaves. However, large amounts of sugars accumulated in plants also inhibit normal photosynthesis and delay plant growth. Pro-Ca promotes sucrose catabolism and reduces sucrose synthesis by increasing leaf SS activity and decreasing SPS activity, alleviating the inhibition of photosynthesis caused by the high concentration of sucrose in the leaves and improving rice photosynthesis. Pro-Ca promotes the transport of sucrose, fructose, and soluble sugars from the source (leaves) to the sink (seeds) and starch synthesis in rice under salt stress. Salt stress severely inhibited rice grouting and significantly reduced the number of solid grains on the primary branches and the number of solid grains on the secondary branches of rice, thereby decreasing the number of solid grains per panicle, 1000-grain weight, fruiting rate, and yield per plant. Pro-Ca mitigated the negative effects of salt stress on the yield components and yield. These results suggest that Pro-Ca improved the effect of salt stress on rice leaf photosynthesis by promoting the metabolism of nonstructural carbohydrates. Pro-Ca promotes the translocation of nonstructural carbohydrates from the source (leaves) to the sink (seeds), promotes the synthesis of starch, and increases the number of solid grains on the branching peduncle and the 1,000-grain weight, which ultimately improves the yield of rice under salt stress.

## 5. Materials and Methods

### 5.1. Plant Materials and Experimental Design

The experiment was conducted in 2023 and 2024 in a daylight greenhouse at the College of Coastal Agricultural Science, Guangdong Ocean University, Zhanjiang, Guangdong, China. Seeds of a salt-sensitive variety, “Huang Huazhan” (HHZ), and a salt-tolerant variety, “Guanghong 3” (GH3), were provided by the College of Coastal Agricultural Science, Guangdong Ocean University. The plant growth regulator for testing was calcium cyclamate (Pro-Ca), and the stock solution (5% Pro-Ca) was provided by Sichuan Runer Technology Co. Full and intact seeds were selected and then sterilized with 3% hydrogen peroxide for 15 min, rinsed several times with distilled water until thoroughly rinsed, soaked in distilled water for 24 h, and germinated in the dark at 30 °C for 24 h. Subsequently, the seeds were sown in seedling trays measuring 54 cm in length and 28 cm in width with holes in the bottom. Seedlings were planted manually when they reached the four-leaf and one-heart stage, and uniformly grown seedlings were selected and transplanted into watertight plastic pots (top diameter 27.5 cm, bottom diameter 18.5 cm, height 23 cm), which were filled with 8 kg of soil, with three holes per pot, two plants per hole, 10 cm hole spacing, and a planting depth of 1.5 cm. The test soil was red soil. The bucket was filled with 3L of water, and the location of the water layer was marked and regularly replenished to maintain the water layer. When the rice panicle is drawn about 1 cm or so, which occurs in more than 30–50% of the panicle in the field, it is known as the breakthrough stage. When the rice seedlings grow naturally to the point where the inverted one-leaf pillows and the inverted two-leaf pillows are 2–3 cm apart (about 5–7 days before the breakthrough stage), a foliar spraying of 100 mg/L Pro-Ca (30 mL of spray per pot) was applied. A total of 24 h after spraying, 80 mmol/L NaCl solution was selected to simulate salt stress, and clear water was used as the control. CK: 0 mM NaCl + 0 mg·L^−1^ Pro-Ca, S: 80 mM NaCl + 0 mg·L^−1^ Pro-Ca, SP: 80 mM NaCl + 100 mg·L^−1^ Pro-Ca. The experiment was conducted in a completely randomized block design with three biological replicates for each treatment. Leaf samples were collected from the whole plant on days 0, 5, 10, 15, 20, 25, and 30 of the filling periods and stored in liquid nitrogen at −40°C in a freezer for physiological and biochemical analyses.

### 5.2. Measurement of Growth Indices

Samples were taken on the 15th day of the filling period, and each treatment was replicated three times. The plant height, root length, internode length, and spike length were measured for each individual plant with a straightedge. The stem base width and inverted internode diameter were measured with vernier calipers. The leaf area was measured with a Yaxm-1241 leaf area meter (Beijing Yaxin Riyi Technology Co., Ltd., Beijing, China). Subsequently, the rice seedlings were dried at 105 °C for 30 min and at 80 °C until they reached a constant weight; then, the aboveground dry weight, root dry weight, and spike dry weight were determined using an electronic balance.

### 5.3. Measurement of Photosynthetic Gas Exchange Parameters

We measured the net photosynthetic rate (Pn), intercellular CO_2_ concentration (Ci), transpiration rate (Tr), and stomatal conductance (Gs) on day 15 of the filling period using the LI-6800 portable photosynthesis measurement system (LI-6800, LI-COR, Lincoln, NE, USA) on a sunny day from 9:00 a.m. to 11:00 a.m., with three replications per treatment [55]. The conditions in the leaf chamber were photosynthetically active radiation (PAR) of 1000 µmol·m^−2^·s^−1^, a CO_2_ concentration of 400 µmol·m^−1^, a leaf temperature of 30 °C, air relative humidity of 70–80%, and an airflow rate of 500 µmol·s^−1^.

### 5.4. Measurement of Photosynthetic Pigment Content

The determination of photosynthetic pigments chlorophyll a (Chl a), chlorophyll b (Chl b), carotenoids (Car), and total chlorophyll (Chl a + b) were determined by Kolomeichuk on the 15th day of the filling period [56]. Fresh leaf (0.1 g) in 10 mL of 95% ethanol was left for 24 h in the dark. The concentrations of chlorophyll a, chlorophyll b, and Car were determined at 665, 649, and 470 nm, respectively.Chlorophyll a (Chl a) = 13.95 D665 − 6.88 D649Chlorophyll b (Chl b) = 24.96 D649 − 7.32 D665Total chlorophyll content = Chl a + Chl bCarotenoids (Car) = (1000 D470 − 2.05 Chl a − 111.48 Chl b)/245

### 5.5. Measurement of Nonstructural Carbohydrate Content

The measurement of carbohydrate content was carried out according to the method proposed by Du et al. [57]. A total of 0.5 g frozen leaf sample was placed in a mortar, ground to powder with 80% ethanol solution (*v*/*v*), and loaded into a centrifuge tube. The tubes were placed in a water bath at 80 °C for 20 min and then centrifuged at 4000 rpm for 5 min. The supernatant was collected and fixed to 25 mL. The remaining residue was extracted three times, as described above. The resulting solution was used to measure the fructose, soluble sugar, and sucrose levels, and the precipitate was used to measure the starch content. The fructose content was measured according to a previous study [58]. For fructose, 0.8 mL of supernatant was mixed with 1.6 mL of 0.1% resorcinol reagent (*w*/*v*) and 0.8 mL of distilled water and heated to 80 °C for 10 min, and then the absorbance was measured at 480 nm. To estimate the sucrose content, 0.4 mL of sugar extract was boiled with 0.2 mL of 2 M NaOH, followed by the addition of 0.8 mL of 0.1% resorcinol and 2.8 mL of 30% HCl. The reaction mixture was incubated for 10 min at 80 °C in a water bath. The absorbance was measured at 480 nm using a spectrophotometer, as described by Du et al. [59]. The starch content was determined using the method of Kuai et al. [60]. After removing the ethanol via evaporation, 2 mL of distilled water was added to the sample, which was then incubated for 15 min at 80 °C. Starch was mixed with 2 mL of 9.2 M HClO_4_ and 2 mL of distilled water. The mixture was centrifuged at 4000 rpm for 10 min, the residue was extracted twice more with 4.6 M HClO_4_ and distilled water, the supernatant was collected, and the volume was fixed to 50 mL. A total of 2.5 mL of the supernatant was mixed with 6.5 mL of anthrone reagent, and the absorbance was measured at 620 nm.

### 5.6. Determination of Key Enzyme Indicators of Sucrose Metabolism

Frozen leaf samples were extracted in 0.1 M of PBS buffer (pH 7.5) containing 5 mM of MgCl_2_, 1 mM of EDTA, 1 mM of EDTA, 0.1% (*v*/*v*) β-mercaptoethanol, and 0.1% (*v*/*v*) Triton X-100 at 4 °C and centrifuged at 10,000 rpm for 15 min at 4 °C. Supernatant was then pipetted into a 10 mL calibration tube. The supernatant was used to determine the activities of sucrose phosphate synthase (SPS) and sucrose synthase (SS). The measurements of SPS (EC 2.4.1.14) and SS (EC 2.4.1.13) were carried out according to the methods proposed by Wongmetha [61] and Baxter [53]. The SPS was assayed in a mixed solution containing 0.1 M of borate buffer (pH 8.0), 15 mM of MgCl_2_, 5 mM of fructose-6-phosphate, 15 mM of glucose-6-phosphate, 10 mM of UDP-glucose, and enzyme extract. The reaction mixture was incubated at 30 °C for 60 min. The reaction was stopped by adding 0.2 mL of 30% KOH, and then the mixture was heated at 100 °C for 10 min. After cooling, anthrone reagent (in H_2_SO_4_) was added, and the absorbance was measured at 620 nm. The SS method is similar to the SPS method, but it contains 0.06 M of fructose instead of fructose-6-phosphate, and it contains no glucose-6-phosphate. The SS and SPS activities were expressed as sucrose (μg)/fresh weight (g)^−1^ min^−1^.

### 5.7. Measurement of Key Enzyme Indicators of Starch Metabolism

Starch phosphorylase (SP EC 2.4.1.7) measurements were performed according to Singh et al. [62]. Frozen leaf samples were homogenized in a buffer containing 100 mM of sodium succinate (pH 5.8), 10% glycerol, 1 mM of EDTA, 15 mM of β-mercaptoethanol, 1 mM of EDTA, and 5 mM of MgCl_2_ and centrifuged at 16,000 rpm for 10 min at 4 °C. Ten ml of supernatant was mixed with 0.8 mL of SDB (100 mM of sodium succinate (pH 5.8), 0.1% bovine serum albumin (*w*/*v*), 10 mM of β-mercaptoethanol, 0.2 mM of EDTA, and 10% glycerol) and 0.1 mL of substrate mixture (100 mM of sodium succinate (pH 5.8), 5% soluble starch (*w*/*v*), 0.1 mM of glucose-1-phosphate 0.2 mM AMP). The mixture was kept at 30 °C for 10 min; then, 2.6 mL of solution (2.6 g of ammonium carbonate in 100 mL of 14% [*v*/*v*] sulfuric acid) and 0.4 mL of solution (0.5% stannous chloride in 0.1 mM of HCl) were added. After shaking at 3000 rpm for 10 min, the absorbance value was measured at 520 nm.

### 5.8. Yield and Yield Components

In 2023 and 2024, eighteen plants of uniform growth were selected from each treatment at maturity for the seed test, with 18 replications to determine the panicle traits and yield components. The panicle length was measured, and the number of primary and secondary branches of the panicle on the main stem, the number of solid grains on the primary branches, the number of empty grains on the primary branches, the number of solid grains on the secondary branches, and the number of empty grains on the secondary branches were recorded. The number of panicles on a single plant was recorded. The weights of solid and empty grains were weighed, and the seed setting rate per plant, 1000-grain weight, and yield per plant were calculated.

## 6. Statistical Analysis

In this experiment, each treatment was repeated three times for each indicator. The mean and standard deviation were calculated from the experimental data. SPSS 25.0 (SPSS Inc., Chicago, IL, USA) was used for the statistical analysis. Repeated variables were compared using Duncan’s multiple range test at a 0.05 level of significance. The data were drawn in Origin 2021 (OriginLab, Northampton, MA, USA). A correlation analysis was calculated and drawn using Origin 2021 (OriginLab, Northampton, MA, USA).

## Figures and Tables

**Figure 1 plants-14-00211-f001:**
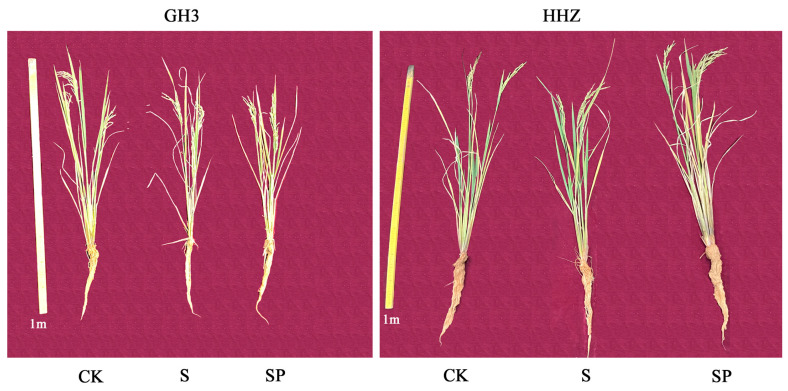
Effect of Pro-Ca on rice phenotype under salt stress. CK: 0 mM NaCl + 0 mg·L^−1^ Pro-Ca, S: 80 mM NaCl + 00 mg·L^−1^ Pro-Ca, SP: 80 mM NaCl + 100 mg·L^−1^ Pro-Ca. “Huang Huazhan” (HHZ). “Guanghong 3” (GH3).

**Figure 2 plants-14-00211-f002:**
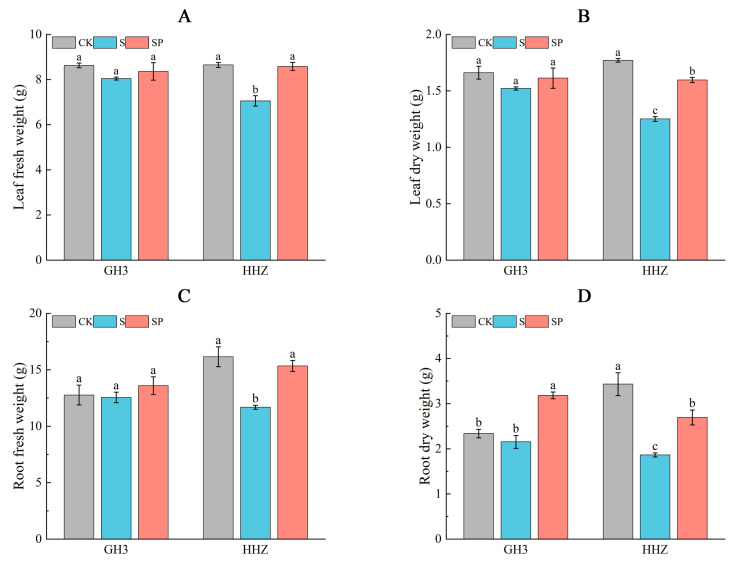
Effect of Pro-Ca on leaf fresh weight (**A**), leaf dry weight (**B**), root fresh weight (**C**), and root dry weight (**D**). CK: 0 mM NaCl + 0 mg·L^−1^ Pro-Ca, S: 80 mM NaCl + 00 mg·L^−1^ Pro-Ca, SP: 80 mM NaCl + 100 mg·L^−1^ Pro-Ca. “Huang Huazhan” (HHZ). “Guanghong 3” (GH3). Values are the mean ± SE of three replicate samples. Different letters in the data column indicate significant differences (*p* < 0.05).

**Figure 3 plants-14-00211-f003:**
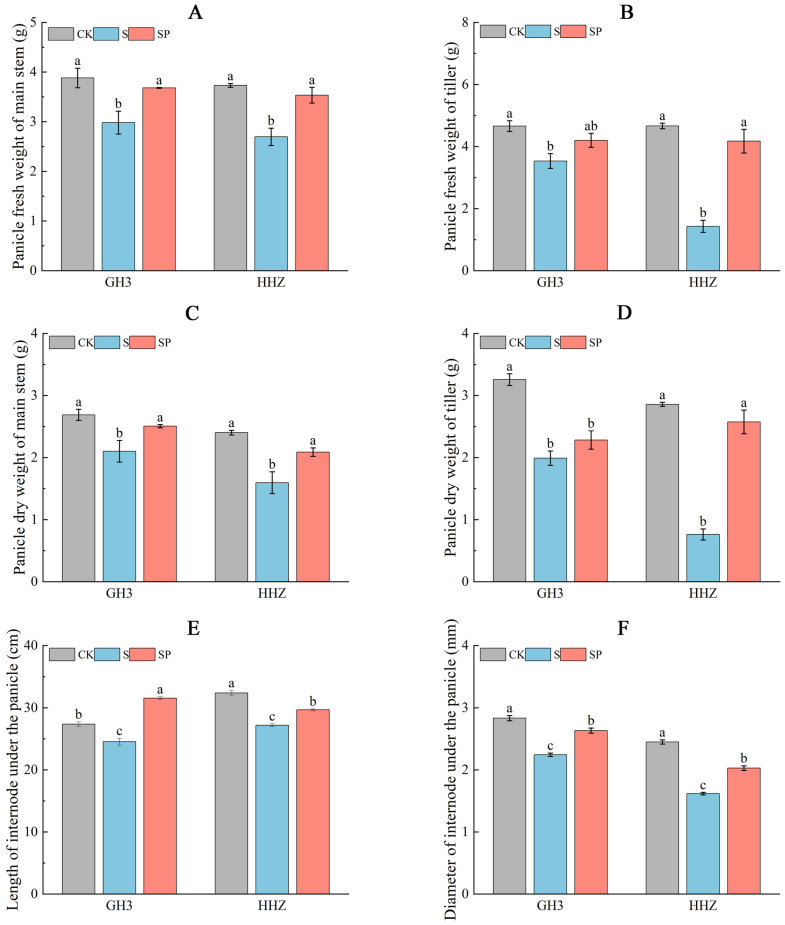
Effect of Pro-Ca on panicle fresh weight of main stem (**A**), panicle fresh weight of tiller (**B**), panicle dry weight of main stem (**C**), panicle dry weight of tiller (**D**), length of internode under the panicle (**E**), and diameter of internode under the panicle (**F**). CK: 0 mM NaCl + 0 mg·L^−1^ Pro-Ca, S: 80 mM NaCl + 00 mg·L^−1^ Pro-Ca, SP: 80 mM NaCl + 100 mg·L^−1^ Pro-Ca. “Huang Huazhan” (HHZ). “Guanghong 3” (GH3). Values are the mean ± SE of three replicate samples. Different letters in the data column indicate significant differences (*p* < 0.05).

**Figure 4 plants-14-00211-f004:**
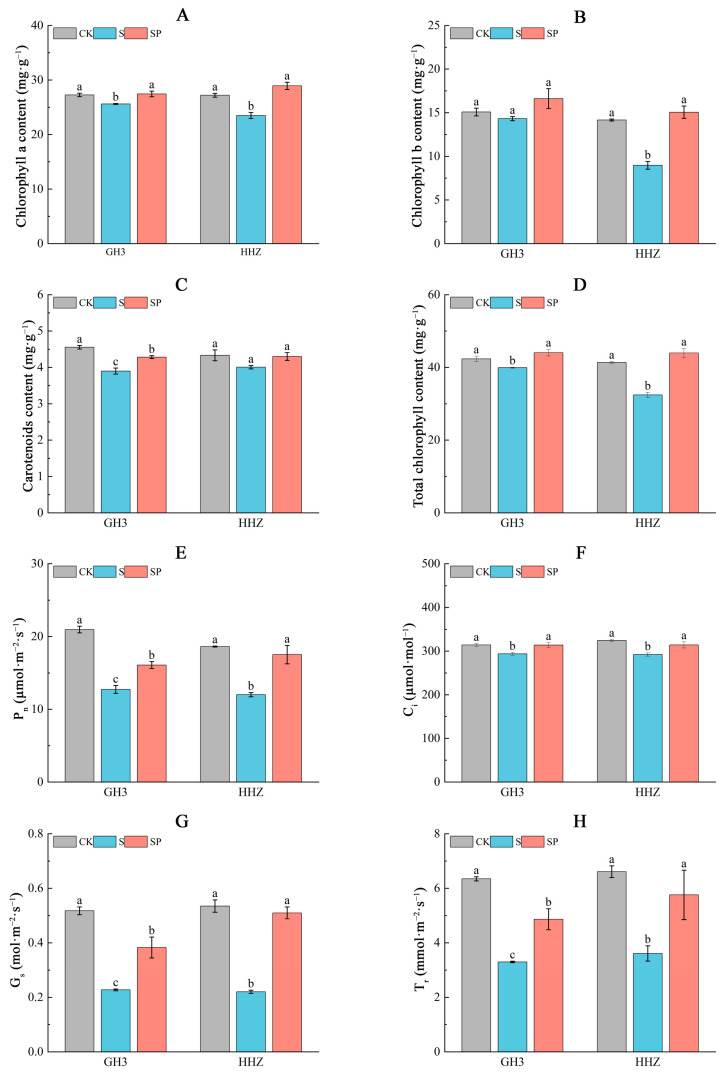
Effect of Pro-Ca on chlorophyll a (**A**), chlorophyll b (**B**), carotenoid (**C**), and total chlorophyll (**D**) contents, net photosynthetic rate (P_n_) (**E**), intercellular CO_2_ concentration (C_i_) (**F**), stomatal conductance (G_s_) (**G**), and transpiration rate (T_r_) (**H**) in HHZ and GH3 under salt stress. CK: 0 mM NaCl + 0 mg·L^−1^ Pro-Ca, S: 80 mM NaCl + 00 mg·L^−1^ Pro-Ca, SP: 80 mM NaCl + 100 mg·L^−1^ Pro-Ca. “Huang Huazhan” (HHZ). “Guanghong 3” (GH3). Values are the mean ± SE of three replicate samples. Different letters in the data column indicate significant differences (*p* < 0.05).

**Figure 5 plants-14-00211-f005:**
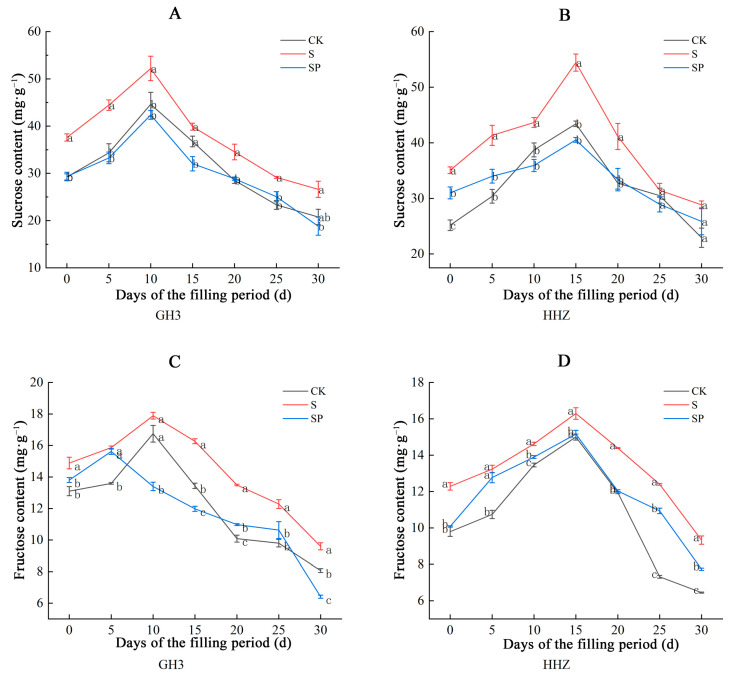
Effect of Pro-Ca on sucrose content (**A**), fructose content (**C**), and soluble sugar content (**E**) in GH3 leaves under salt stress. Effect of Pro-Ca on sucrose content (**B**), fructose content (**D**), and soluble sugar content (**F**) in HHZ leaves under salt stress. CK: 0 mM NaCl + 0 mg·L^−1^ Pro-Ca, S: 80 mM NaCl + 00 mg·L^−1^ Pro-Ca, SP: 80 mM NaCl + 100 mg·L^−1^ Pro-Ca. “Huang Huazhan” (HHZ). “Guanghong 3” (GH3). Values are the mean ± SE of three replicate samples. Different letters in the data column indicate significant differences (*p* < 0.05).

**Figure 6 plants-14-00211-f006:**
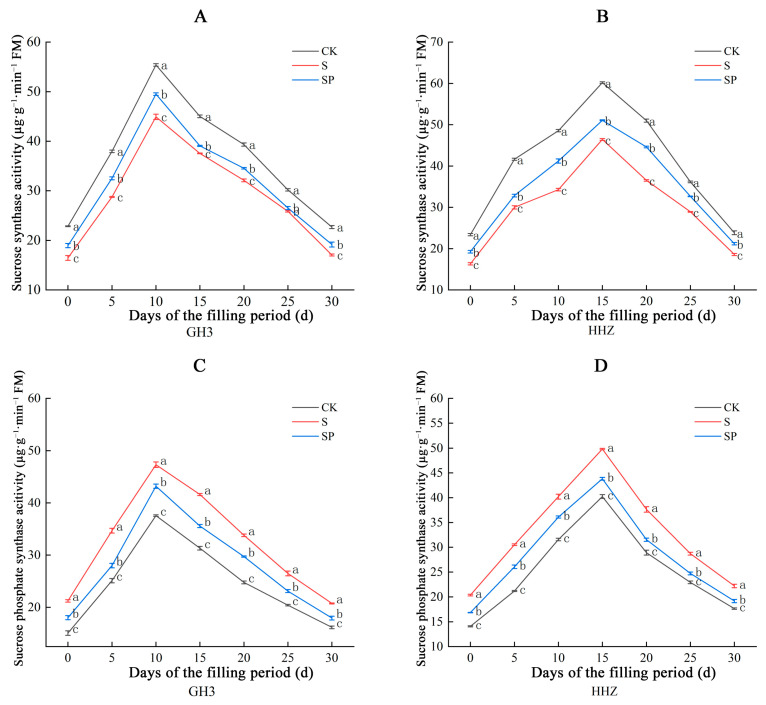
Effect of Pro-Ca on sucrose synthase activity (SS) (**A**) and sucrose phosphate synthase activity (SPS) (**C**) on GH3 leaves under salt stress. Effect of Pro-Ca on sucrose synthase activity (**B**) and sucrose phosphate synthase activity (**D**) on HHZ leaves under salt stress. CK: 0 mM NaCl + 0 mg·L^−1^ Pro-Ca, S: 80 mM NaCl + 00 mg·L^−1^ Pro-Ca, SP: 80 mM NaCl + 100 mg·L^−1^ Pro-Ca. “Huang Huazhan” (HHZ). “Guanghong 3” (GH3). Values are the mean ± SE of three replicate samples. Different letters in the data column indicate significant differences (*p* < 0.05).

**Figure 7 plants-14-00211-f007:**
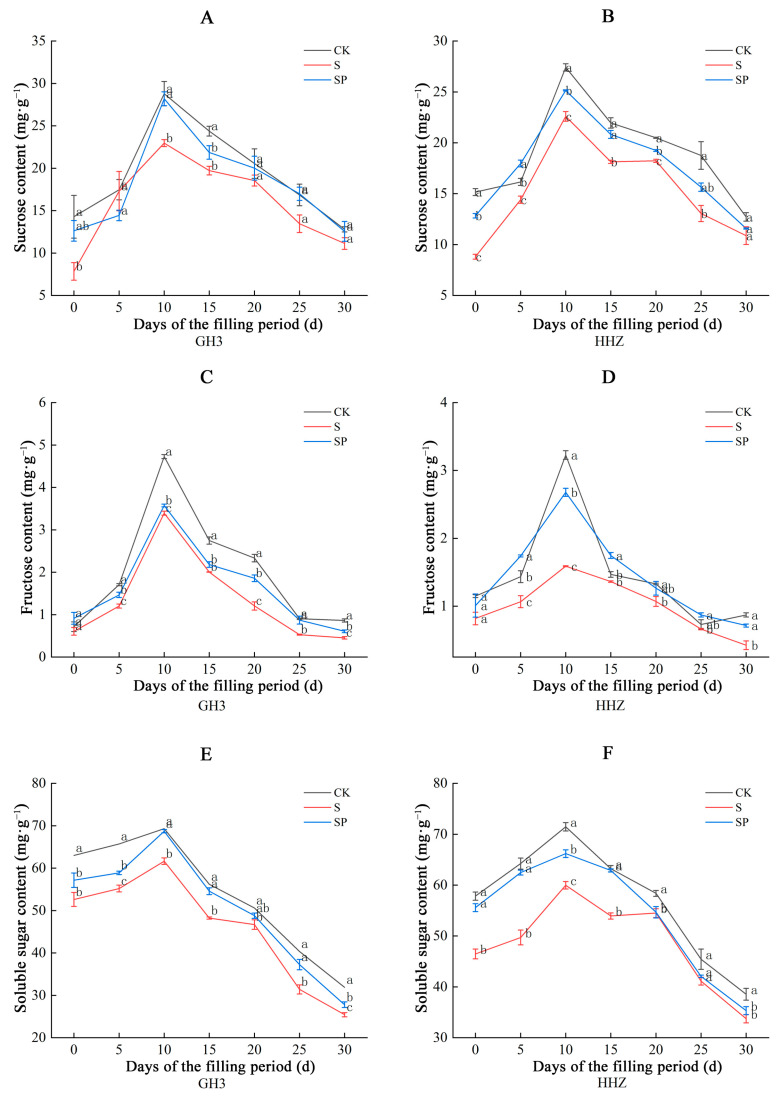
Effect of Pro-Ca on sucrose content (**A**), fructose content (**C**), and soluble sugar content (**E**) in GH3 seeds under salt stress. Effect of Pro-Ca on sucrose content (**B**), fructose content (**D**), and soluble sugar content (**F**) in HHZ seeds under salt stress. CK: 0 mM NaCl + 0 mg·L^−1^ Pro-Ca, S: 80 mM NaCl + 00 mg·L^−1^ Pro-Ca, SP: 80 mM NaCl + 100 mg·L^−1^ Pro-Ca. “Huang Huazhan” (HHZ). “Guanghong 3” (GH3). Values are the mean ± SE of three replicate samples. Different letters in the data column indicate significant differences (*p* < 0.05).

**Figure 8 plants-14-00211-f008:**
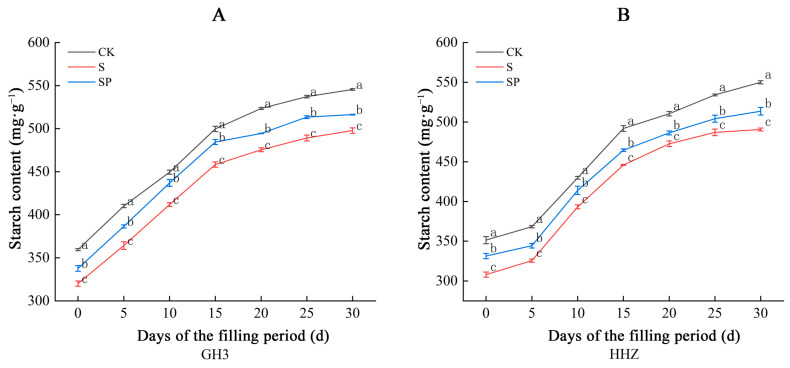
Effect of Pro-Ca on starch content in GH3 (**A**) and HHZ (**B**) seeds under salt stress. CK: 0 mM NaCl + 0 mg·L^−1^ Pro-Ca, S: 80 mM NaCl + 00 mg·L^−1^ Pro-Ca, SP: 80 mM NaCl + 100 mg·L^−1^ Pro-Ca. “Huang Huazhan” (HHZ). “Guanghong 3” (GH3). Values are the mean ± SE of three replicate samples. Different letters in the data column indicate significant differences (*p* < 0.05).

**Figure 9 plants-14-00211-f009:**
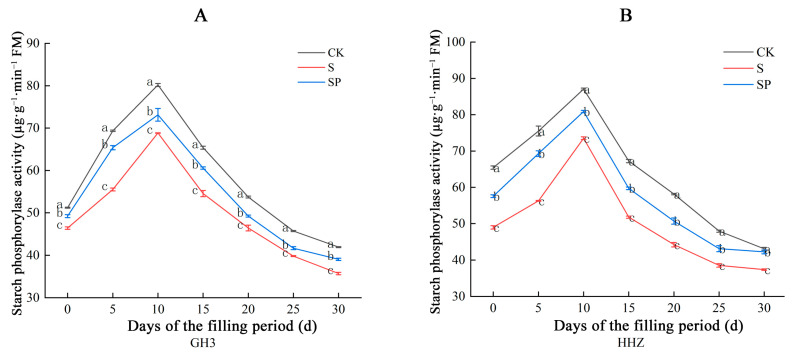
Effect of Pro-Ca on starch phosphorylase activity in GH3 (**A**) and HHZ (**B**) seeds under salt stress. CK: 0 mM NaCl + 0 mg·L^−1^ Pro-Ca, S: 80 mM NaCl + 00 mg·L^−1^ Pro-Ca, SP: 80 mM NaCl + 100 mg·L^−1^ Pro-Ca. “Huang Huazhan” (HHZ). “Guanghong 3” (GH3). Values are the mean ± SE of three replicate samples. Different letters in the data column indicate significant differences (*p* < 0.05).

**Figure 10 plants-14-00211-f010:**
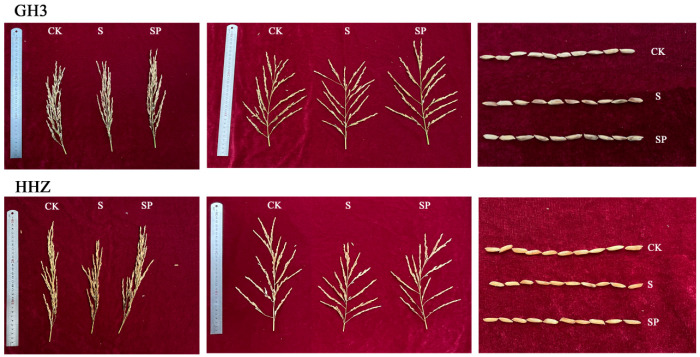
Effect of Pro-Ca on GH3 and HHZ panicle phenotypes under salt stress. CK: 0 mM NaCl + 0 mg·L^−1^ Pro-Ca, S: 80 mM NaCl + 00 mg·L^−1^ Pro-Ca, SP: 80 mM NaCl + 100 mg·L^−1^ Pro-Ca. “Huang Huazhan” (HHZ). “Guanghong 3” (GH3).

**Figure 11 plants-14-00211-f011:**
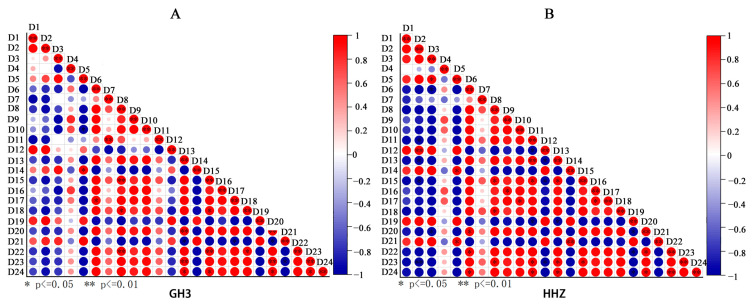
For GH3 (**A**) and HHZ (**B**), D1 (leaf sucrose content), D2 (leaf fructose content), D3 (leaf soluble sugar content), D4 (leaf SS activity), D5 (leaf SPS activity), D6 (seed sucrose content), D7 (seed fructose content), D8 (seed soluble sugar content), D9 (seed amylose content), D10 (seed starch phosphorylase activity), D11 (2023 number of solid grains on primary branches), D12 (2023 number of empty grains on primary branches), D13 (2023 number of solid grains on secondary branches), D14 (2023 number of empty grains on secondary branches), D15 (2023 1000-grain weight), D16 (2023 seed setting rate), D17 (2023 yield per plant), D18 (2024 number of solid grains on primary branches), D18 (2024 number of solid grains on primary branches), D19 (2024 number of solid grains on secondary branches), D19 (2024 empty grains on primary branches), D20 (2024 solid grains on secondary branches), D21 (2024 empty grains on secondary peduncle), D22 (2024 1000-grain weight), D23 (2024 seed setting rate), and D24 (2024 yield per plant) indices were analyzed for correlation. We selected nonstructural carbohydrates and enzyme activities on day 15 of the filling period for correlation analysis.

**Table 1 plants-14-00211-t001:** Effects of Pro-Ca on rice yield components under salt stress.

Year	Variety	Treatment	Panicle Length (cm)	Number of Primary Branches	Number of Secondary Branches	Number of Solid Grains on Primary Branches	Number of Empty Grains on Primary Branches	Number of Solid Grains on Secondary Branches	Number of Empty Grains on Secondary Branches
2023	GH3	CK	22.22 ± 0.26 a	12.17 ± 0.20 a	43.67 ± 1.33 a	52.06 ± 2.02 b	16.94 ± 2.05 a b	94.56 ± 3.08 a	37.44 ± 3.92 a
		S	20.97 ± 0.25 b	12.39 ± 0.27 a	43.61 ± 1.13 a	48.94 ± 1.79 b	22.22 ± 2.28 a	68.17 ± 3.44 b	60.94 ± 2.77 b
		SP	22.18 ± 0.25 a	12.67 ± 0.16 a	45.67 ± 0.81 a	59.94 ± 1.65 a	12.39 ± 1.22 b	90.39 ± 3.08 a	46.06 ± 4.47 a
	HHZ	CK	24.13 ± 0.29 a	12.22 ± 0.19 a	38.33 ± 1.22 a	55.89 ± 2.05 a	11.11 ± 1.11 b	88.72 ± 3.07 a	27.56 ± 2.88 c
		S	22.13 ± 0.38 b	12.06 ± 0.17 a	37.39 ± 0.65 a	39.94 ± 1.50 b	21.11 ± 1.29 a	50.28 ± 1.76 b	59.06 ± 2.20 a
		SP	24.36 ± 0.14 a	12.11 ± 0.18 a	39.61 ± 0.91 a	53.33 ± 2.02 a	12.50 ± 1.08 b	82.33 ± 2.45 a	40.56 ± 2.78 b
2024	GH3	CK	22.56 ± 0.58 a	12.56 ± 0.44 a	33.11 ± 1.82 a	44.11 ± 2.47 a	26.83 ± 1.21 b	69.22 ± 5.05 a	50.39 ± 2.48 a
		S	20.83 ± 0.52 b	12.06 ± 0.45 a	32.06 ± 1.65 a	39.11 ± 2.04 a	33.50 ± 1.78 a	50.11 ± 2.32 b	68.72 ± 2.48 b
		SP	22.29 ± 0.40 a	12.17 ± 0.33 a	34.94 ± 1.46 a	44.11 ± 1.71 a	26.17 ± 1.81 b	66.00 ± 3.40 a	54.17 ± 1.91 a
	HHZ	CK	23.93 ± 0.46 a	13.00 ± 0.40 a	33.28 ± 2.30 a	48.78 ± 2.98 a	20.22 ± 1.79 b	66.50 ± 4.68 a	36.94 ± 3.25 c
		S	22.23 ± 0.32 b	12.39 ± 0.32 a	31.00 ± 1.09 a	35.94 ± 2.63 b	34.17 ± 2.77 a	30.94 ± 2.64 b	64.61 ± 2.31 a
		SP	23.87 ± 0.45 a	13.11 ± 0.28 a	34.28 ± 1.26 a	48.78 ± 2.21 a	23.28 ± 1.82 b	56.89 ± 2.99 a	48.44 ± 1.83 b

Note: Values are expressed as the means ± SE of 18 replicates, and different letters after the data in the same column indicate significant differences between treatments (*p* < 0.05). CK: 0 mM NaCl + 0 mg·L−1 Pro-Ca, S: 80 mM NaCl + 00 mg·L−1 Pro-Ca, SP: 80 mM NaCl + 100 mg·L−1 Pro-Ca. “Huang Huazhan” (HHZ). “Guanghong 3” (GH3).

**Table 2 plants-14-00211-t002:** Effects of Pro-Ca on rice yield and its components under salt stress.

Year	Variety	Treatment	The Productive Panicle Number per Pot	Filled Grain Number per Panicle	Number of Grains per Panicle	1000-Grain Weight (g)	Seed Setting Rate	Grain Yield per Pot (g)
2023	GH3	CK	3.39 ± 0.12 a	107.23 ± 1.85 a	201.00 ± 4.84 a	20.45 ± 0.20 a	0.72 ± 0.01 a	7.41 ± 0.27 a
		S	3.06 ± 0.15 a	79.07 ± 3.32 b	186.39 ± 3.85 b	18.08 ± 0.23 b	0.59 ± 0.01 c	4.29 ± 0.21 c
		SP	3.00 ± 0.16 a	111.76 ± 4.32 a	208.78 ± 4.73 a	20.34 ± 0.29 a	0.66 ± 0.01 b	6.59 ± 0.16 b
	HHZ	CK	3.11 ± 0.18 a	111.01 ± 3.75 a	183.28 ± 4.68 b	19.82 ± 0.27 a	0.75 ± 0.01 a	6.68 ± 0.30 a
		S	3.00 ± 0.21 a	63.04 ± 2.21 b	160.22 ± 3.56 c	17.02 ± 0.49 b	0.53 ± 0.02 c	3.18 ± 0.25 c
		SP	2.94 ± 0.13 a	86.94 ± 3.50 a	195.67 ± 4.08 a	19.57 ± 0.27 a	0.63 ± 0.01 b	4.93 ± 0.21 b
2024	GH3	CK	2.33 ± 0.14 a	100.85 ± 2.64 a	190.56 ± 6.77 a	26.10 ± 0.83 a	0.62 ± 0.01 a	5.87 ± 0.21 a
		S	2.28 ± 0.11 a	86.59 ± 3.31 b	182.56 ± 6.75 a	22.84 ± 0.69 b	0.51 ± 0.01 b	4.38 ± 0.15 b
		SP	2.33 ± 0.11 a	98.24 ± 2.96 a	193.22 ± 5.14 a	26.01 ± 1.1 a	0.60 ± 0.01 a	5.69 ± 0.28 a
	HHZ	CK	2.28 ± 0.11 a	104.25 ± 2.31 a	172.17 ± 7.30 a	26.05 ± 1.07 a	0.64 ± 0.02 a	5.92 ± 0.23 a
		S	2.22 ± 0.10 a	71.29 ± 2.49 c	150.11 ± 6.64 b	21.38 ± 0.45 b	0.50 ± 0.02 b	3.33 ± 0.13 b
		SP	2.17 ± 0.09 a	96.03 ± 2.62 b	170.72 ± 5.47 a	24.91 ± 1.05 a	0.62 ± 0.01 a	5.31 ± 0.35 a

Note: Values are expressed as the means ± SE of 18 replicates, and different letters after the data in the same column indicate significant differences between treatments (*p* < 0.05). CK: 0 mM NaCl + 0 mg·L−1 Pro-Ca, S: 80 mM NaCl + 00 mg·L−1 Pro-Ca, SP: 80 mM NaCl + 100 mg·L−1 Pro-Ca. “Huang Huazhan” (HHZ). “Guanghong 3” (GH3).

## Data Availability

The data presented in this study are available on request from the corresponding author.

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
