# Peer review of "Prohexadione Calcium Improves Rice Yield Under Salt Stress by Regulating Source–Sink Relationships During the Filling Period"

_plants, 2025, doi:10.3390/plants14020211_

Round 1

Reviewer 1 Report

Comments and Suggestions for Authors

This paper reported the effects of spraying Prohexadione (Pro-Ca) to rice at the pre-heading stage on mitigating salt-induced reduction in yield of a salt-tolerant and a salt-sensitive rice cultivar focusing on the role of Pro-Ca in modulating source-sink relations. The paper presented an interesting set of data. The experimental plans and methodology were adequately described, and the interpretation and conclusion were aligned with the results. However, there are some important discrepancies in the Results section which need to be clarified as follows.

1.       The authors presented the yield data for 2023 and 2024 (Table 1 and 2), but stated that the experiment was conducted in 2022 in the M &M.

2.       If the experiments were conducted in 2 years, why only one set of data on the leaf physiology (photosynthesis, carbohydrates, enzyme activity etc.) were presented?

3.       Description of results for each Figure is too lengthy, the authors should try to shorten it without loosing important messages.

4.       The method for correlation analysis was not presented in 6. Statistical Analysis

5.       Figures 4, 6, 7, 8 – X-axis of the graphs lacks description

6.       There are some typing errors, for example;

6.1) Line 150, 154 and 156 – the authors wrote ‘IR29 and HD96-1’ instead of ‘GH3 and HHZ’

6.2) Line 24 – ‘sucrose phosphorus activity’ must be corrected to ‘phosphate’

6.3) Line 543 – ‘the early stage of irrigation??’

6.4) Line 643 – ‘protect the ability of rice to irrigate???’

6.5) Line 116 – ‘In salt stress inhibited..” should be ‘Salt stress inhibited…”

6.6) Line 776 – ‘.panicle….’ should be ‘Panicle…’

6.7) Line 152 – ‘figure’

6.8) Table 2 – add the unit (g) after 1000-grain weight and grain yield per pot

Reviewer 2 Report

Comments and Suggestions for Authors

It is a well-planned experiment and the manuscript is well-written and well-presented. It can be published after minor revisions:

  1. L716 & 718: (Chl a + B) “B” should be b.

  2. L715: Authors should cite papers for “Measurement of Photosynthetic Pigment Content” section. 

  3. There are many superscript and subscript issues in the manuscript. Please revise these problems.

  4. L755: “enzyme extracts Perform the assay.” sentence not clear.

  5. The scale in the figure 1 is not clear… 

  6. Recheck the legend of the figure 1. The last sentenc is not applicable for this figure.

  7. Legend of all figures should contain the full form of the abbreviation of genotypes.

  8. L495: What is “nutrient growth”. It is confusing?

Reviewer 3 Report

Comments and Suggestions for Authors

This manuscript presents the effects of prohexadione calcium application to salt tolerance of rice especially focusing on photosynthesis ability and sink-source balance. The referee believes that this manuscript provides interesting scientific information on the crop management under salinity stress, and has merit for publication in Plants; however, it also has several deficiencies that need to be addressed before it can be accepted for publication. 

/Description of the ‘Results’ is so redundant. Summarize more concisely with keeping the points.

/How about the differences of source size among plants?

/Line 116: ‘On day 15 of the filling stage, the S treatment significantly decreased the panicle fresh weight of main stem by 23.20% and 27.61%,’

I could not read the text clearly. Check that what is written is correct.

/’rice species’ -> ‘rice varieties’

/For which growth stage (filling period) of leaves and grains, were the sugar content data used in the correlation analysis taken?

/Writing of the ‘Discussion 3.4’ is also redundant, and most of the contents there is duplicate of the ‘Results’. Summarize the essential point.

/In the ‘Discussion’, you commented that almost all of the results were ‘similar to the results of a previous study’. What are the novel findings in this study?

/Line 586: ‘by inhibiting the transport of sucrose’

Is the salt stress preventing the sugar transport?, or are the plants actively retaining sugars in the leaves?

/Both tolerant and susceptible varieties were used, but the discussion has not been given to the relationship between the degree of salinity tolerance and the efficacy of Pro-Ca (only the results are shown).

/In this study, the P-treatment (no-NaCl, Pro-Ca+) was not set up. What would the sugar transfer syndrome (sink-source balance) be changed, if you apply Pro-Ca to CK plant? Is Pro-Ca effective only under conditions of intense stress?

/It is known that the alleviation effects of Pro-Ca observed in this study can also be obtained if we use CaCl2 or other Ca-synthesis as a Ca source. What is the significance of using Pro-Ca?

/How much amount of 100 mg/L Pro-Ca solution you splayed per pot?

Reviewer 4 Report

Comments and Suggestions for Authors

The manuscript "Prohexadione Calcium Improves Rice Yield Under Salt Stress by Regulating Source-Sink Relationships During the Filling Period" is very interesting.
Some points for attention from the authors:

1. Line 90: Define the Fv/Fm and Fv/Fo relationships.

2. Line 130: Figure 1: Remove the letter "A" above the photographic image. Figure 1 is not a graph, so the caption "Values ​​are the mean ± SE of three replicate samples. Different letters in the data column indicate significant differences (p < 0.05)" is not required and should be removed.

3. Line 147: The description of lines 144-147 does not correspond to figure 4.

4. Line 167: The description of lines 166-167 does not correspond to figure 3.

5. Line 210: The x-axes of the graphs in Figure 4 must indicate the unit of measurement (time in days).

6. Line 248: Figure 5 is not cited in the main text.
The x-axes of the graphs in Figure 5 must indicate the unit of measurement (time in days).

7. Line 257: The description of lines 255-270 does not correspond to Fig. 3.

8. Line 298: Figure 6 is not cited in the main text. The graphs in Figure 6 must include units of measurement on the x-axes.

9. Figure 7 should be cited in the text before its appearance.
The graphs in Figure 7 must include units of measurement on the x-axes.

10. The graphs in Figure 8 must include units of measurement on the x-axes.

11. The phrase "Values ​​are the mean ± SE of three replicate samples. Different letters in the data column indicate significant differences (p < 0.05)" should be removed from the caption of Figure 9.

12. The presentation of Table 1 needs to be improved.

13. Line 453: Figure 7 does not correspond to the Pearson correlation analysis.

14. Figure 10 is not cited in the main text.

15. Lines 692-693, 713-714: Correct the measurement units, some ​​must be in superscript.

16. Line 758: Correct the chemical formula of sulfuric acid.

17. Review units of measurement for SS and SPS activities. "G" should be "g".

Round 2

Reviewer 1 Report

Comments and Suggestions for Authors

The authors have thoroughly revised the manuscript following the reviewers' comments. I have no further comments. 

Author Response

Thank you for your patience. This comment should be sent to the editor to read, it shouldn't be for the author.

Reviewer 3 Report

Comments and Suggestions for Authors

/Line 118: omit ‘also’

/Line 155: ‘IR29 and HD96-1’ Check that what is written is correct.

/You collected 7 data for sucrose, fructose and total soluble sugars contents in leaves and grains, respectively, in the different days of the filling period. Which data you used for correlation analysis?

/In the last review comments;

“Comments 2: /How about the differences of source size among plants?

Response 2:Thanks for valuable suggestions. Agreed, revised the appropriate details. Mention exactly where in the revised manuscript this change can be found – line546-549.

I asked ‘How about the difference of plant body mass among treated plants?’ You just compared the mass of the roots and panicles, but information on the mass of the photosynthetic organs should be required for an accurate discussion.

/Line 513: What is ‘weak grains’?

/Line 519: ‘bank’ -> ‘sink’

/Line 677: ‘2.5 10 ml’ Check that what is written is correct.

/Line 723: ‘Origin 2021’ Write according to the instruction like the line 720.
